# CRISPR/Cas9-loaded stealth liposomes effectively cleared established HPV16-driven tumours in syngeneic mice

Luqman Jubair[1,2], Alfred K. Lam[3], Sora Fallaha[1], Nigel A. J. McMillan[1,2]*

**1** School of Medical Sciences, Griffith University, Gold Coast, Queensland, Australia, **2** Menzies Health Institute Queensland, Queensland, Australia, **3** School of Medicine, Griffith University, Gold Coast, Queensland, Australia

* n.mcmillan@griffith.edu.au

**Data Availability Statement:** All relevant data are within the manuscript and its Supporting Information files.

## Abstract

Gene-editing has raised the possibility of being able to treat or cure cancers, but key challenges remain, including efficient delivery, *in vivo* efficacy, and its safety profile. Ideal targets for cancer therapy are oncogenes, that when edited, cause cell death. Here, we show, using the human papillomavirus (HPV) type 16 cancer cell line TC1, that CRISPR/Cas9 targeting the E7 oncogene and packaged in PEGylated liposomes cleared established tumours in immunocompetent mice. Treatment caused no significant toxicity in the spleen or liver. An ideal therapeutic outcome would be the induction of an immunogenic cell death (ICD), such that recurrent tumours would be eliminated by the host immune system. We show here for the first time that CRISPR/Cas9-mediated cell death via targeting E7 did not result in ICD. Overall, our data show that *in vivo* CRISPR/Cas targeting of oncogenes is an effective treatment approach for cancer.

## Introduction

While gene therapy has long held a promise in treating a range of diseases, the field has been beset by issues of efficacy, immunogenicity and rare activation of oncogenes [1]. The discovery of siRNAs further refined our ability to treat diseases, yet it took 20 years before the first siRNA-based therapy, Onpattro™, was approved in 2018 [2]. Cas9/guide RNA (gRNA) technology, derived from the CRISPR/Cas9 bacterial immune system, is poised to revolutionize medicine via its ability to correct disease-causing genes, particularly in the cancer setting where the driver oncogenes are known. However, several challenges remain ahead of its clinical translation, such as the targeting specificity, the delivery of the CRISPR/Cas9, the immunogenicity of the delivery vehicle and CRISPR/Cas9 components, and whether the latter would affect the *in vivo* treatment efficacy [3]. In addition, treating early-stage cancers, when they are still confined to the primary site or organ, is surgically possible with a high success rate. Once metastasized, however, the treatment becomes more challenging as the systemic delivery of cancer therapies has proven to be difficult, with significant side-effects and poor overall efficacy [4]. There is a clear need to optimize systemic delivery vehicles to deliver targeted therapeutics to the desired sites, which should be safer and more effective.

**Funding:** The funders had no role in study design, data collection and analysis, decision to publish, or preparation of the manuscript.

**Competing interests:** The authors have declared that no competing interests exist.

Many viral and non-viral delivery systems have been tested as a modality for the systemic delivery of CRISPR/Cas9 with varying success [5]. In immunocompromised mice, we previously demonstrated that packaging Cas9/gRNA plasmids in PEGylated liposomes via the hydration of freeze-dried matrix (HFDM) could effectively deliver payloads to cervical cancer xenografts [6]–a disease characterized by its addiction on the expression of human papillomavirus (HPV) oncogenes, E6 and E7 [7]. Beyond the targeting and the delivery of treatment, the immunogenicity of CRISPR/Cas9 components could be a hurdle as the introduction of nucleic acids/proteins may elicit innate, cellular, and humoral immune responses [3]. Indeed, Cas9 exposure was previously shown to activate Cas9-specific IgM and IgG antibodies in mice [8], which may neutralize the edited cells in the long-term. In immunocompetent mice, we have previously shown that shielding liposomes with PEGylation, a non-toxic and non-immunogenic polyether diol layer, could protect siRNAs and evade the immune system with no significant immune activation [9,10]. However, the systemic toxicity of PEGylated liposomes loaded with CRISPR/Cas9 is yet to be explored in immunocompetent mouse models.

A further challenge is post-editing immunity as the generation of random indels may give rise to novel antigens that are immunologically foreign. In previous work, we showed that the intravenously administered stealth liposomes coating Cas9/gRNAs targeting HPV16E7 (16E7) or HPV18E7 (18E7) oncogenes effectively eliminated established CasKi (HPV16 +ve) or HeLa (HPV18 +ve) tumours in immunocompromised mice [6]. However, whether this effect would be sustained under competent immune conditions remains unknown. Finally, to improve the long-term anticancer effect of treatment, eliciting oxidative ER stress is essential for developing immunity against cancerous cells through emission of damage-associated molecular patterns (DAMPs) [11]. This allows the dying cancer cells to induce host anticancer immunity, a phenomenon called immunogenic cell death (ICD) [12]. ICD requires the surface exposure of intracellular chaperones such as calreticulin (CALR), heat shock protein 90 (HSP90) or HSP70, ATP release, and high-mobility group box-1 (HMGB1) protein release [11]. This would prevent cancer reoccurrence and to date this has not been tested in CRISPR/Cas therapies.

Here we test PEGylated liposomes containing Cas9/16E7-expressing plasmids for their ability to target and eliminate established HPV16 E7-driven tumours in syngeneic mice, as well as their toxicity profile. We also examine, for the first time, whether CRISPR/Cas-mediated cell death can result in ICD and thereby prevent cancer reoccurrence.

## Methods and materials

### Cell culture, transfection and plasmid

CasKi (HPV 16 +ve, passage number 6), C33A (HPV -ve, passage 12), HeLa (passage number 23), and Jurkat cell lines (purchased from American Type Culture Collection) were cultured in Dulbecco's Modified Eagle's Medium (DMEM) supplemented with 10% heat-inactivated fetal bovine serum (FBS), and 1% antibiotic mixture of penicillin G, streptomycin sulfate and L-Glutamine. TC1 cell line (passage number 14, derived from primary lung epithelial cells of C57BL/6, complemented with HPV 16E7 gene, were kindly provided by Prof James Wells, Translational Research Institute, Diamantina Institute, The University of Queensland, Brisbane, Australia) was cultured in RPMI 1640, supplemented with 10% heat inactivated FBS, insulin, 2 mM L-glutamine, 1 mM Pyruvate, 0.1 mM minimal essential medium with nonessential amino acids, penicillin 100U/ml and 100 μg streptomycin/ml. Mycoplasma testing by PCR was carried out on monthly basis to ensure the cell lines are contamination-free. All cell lines were authenticated by short tandem repeats (STR) analysis at the Griffith DNA Sequencing Facility in accordance with ATCC guidelines.

The Cas9 and gRNA expressing plasmid was purchased from Addgene (px330S-2, #58778). The target site within the HPV16E7 gene was selected using CRISPRDirect online tool [13]. The 16E7 gRNA (target sequence: <u>ccg</u>gacagagcccattacaatat) and control (non-specific) gRNA (target sequence: tcgtactctacagcagatgc) were cloned into px330S-2 plasmid as described elsewhere [14]. Lipofectamine 3000 reagent (Thermo Fisher Scientific) was used according to the manufacturer's instructions

## Cell viability assay

The MTT (3-(4,5-dimethylthiazol-2-yl)-2,5-diphenyltetrazolium bromide) tetrazolium reduction assay was used to determine the effect of 16E7 targeting on cell viability. Three days after treatment, 50 μL of MTT (12 mM) was added to a fresh 450 μL of DMEM per well (24-well plate) and incubated at 37°C for four hours. The development of the blue formazan due to MTT metabolism by viable cells was assessed by quantifying its optical density at a wavelength of 544 nm.

## ATP release assay

The release of ATP after treatment was assessed using colorimetric ATP assay kit (Abcam, #83355) according to manufacturer's protocol. Briefly, TC1 cells were transfected with either Cas9+nonspecific gRNA, or Cas9+ 16E7 gRNA or untreated in triplicates. Samples were incubated at room temperature for 30 minutes in a dark room. Optical density was measured at 570 nm.

## T7E1 assay

The editing efficiency was estimated by the T7E1 assay as described elsewhere [15]. After treatment, the target 16E7 gene was amplified by PCR using two primers flanking the editing site. The digested products were analysed by 2% agarose gel electrophoresis.

## Western blot

The effect of 16E7 targeting was assessed by quantifying retinoblastoma (Rb) protein via western blotting. TC1 cells were seeded in T25 flask and transfected, 48 hours after transfection, cells were treated with MG132 (20 μM) for 12 hours, then lysed with RIPA buffer and halt protease inhibitor. Samples were loaded into 12% SDS-PAGE gel, for three hours (at 120 V, 4°C). Membranes were probed with primary antibody overnight with agitation at 4°C. Rb antibody (BD Sciences, #610261), HMGB-1 (abcam, ab18256). Jurkat and HeLa whole cell lysates were used as a positive control for Rb and HMGB-1 antibodies, respectively.

## In vivo testing

For *in vivo* testing, $1 \times 10^5$ TC1 cells were suspended in PBS then subcutaneously injected into immunocompetent C57BL/6J mice (six mice per test group, 6-week of age, purchased from the University of Queensland Biological Resources, Brisbane, Australia). The animals were sheltered in the animal facility unit at Griffith University and care was provided by trained staff (divided into five mice per group, each group was kept in a conventional cage, normal bedding, normal feeding and watering, ambient humidity and temperature). Cas9 and 16E7 gRNA expressing plasmids were packaged into PEGylated liposomes by the hydration of freeze-dried matrix (HFDM) [16]. The packaging capacity and the PEGylation ratio were determined using procedures described previously [16]. Once tumours were ≈ 50 mm³ in size, ten microgram/dose of either treatment (Cas9+16E7 targeting gRNAs), control (Cas9

+nonspecific gRNA), or PBS were injected via the tail vein at different time points. The well-being of the animals was assessed on daily basis following Griffith University Guidelines for Animal Care and Use (general health assessment including eating, locomotion, behaviour, appearance, and weight loss). To minimise animal stress, Isoflurance anaesthesia was used prior to inoculation of tumour cells or the assessment of tumour volume. No animal death was reported prior to reaching the humane endpoint. Tumours were monitored on daily basis after inoculation for a total of 38 days until the conclusion of the experiment. As tumours appeared to take mostly spheroid geometry, tumour volume was assessed using the following formula: $V = (W(2) \times L)/2$, where V represents the volume, W represents the width, and L represents the length.

From previous work, the number of treatments required varied based on the growth rate of the cell line. TC1 cells are rapidly growing tumour cells, and therefore we tested two treatment arms; the first group received a total of seven injections of treatment (Cas9+16E7 gRNA) on second-daily basis, followed by four injections of control (Cas9+nonspecific gRNA). The second group continued treatment (Cas9+16E7) until the experimental endpoint was reached (tumour volume of 1000 mm$^3$, culling by carbon dioxide method, or cervical dislocation as a secondary method of euthanasia). Tissue blocks were collected from mice' tumour, liver, and spleen for immunohistochemical staining. Haematoxylin and eosin (H&E) stained sections were cut from the tissue blocks to examine tissue morphology, inflammation, and pathological alterations. Cleaved Caspase-3 Rabbit monoclonal antibody (Cell Signaling, #9664) was used to assess apoptosis.

Immunogenic cell death was assessed by injecting $1 \times 10^6$ of pre-treated TC1 cells with Cas9 +16E7 treatment (2.5 µg of total DNA), cisplatin treated cells (50 µM for six hours), or Mitoxantrone (1 µM for six hours) into the left flank of C57BL/6J mice (five mice per group). Seven days later, a rechallenge experiment was undertaken by injecting $1 \times 10^5$ of viable TC1 cells into the right flank of these mice. This project has been approved by Griffith University Ethics Committee (project number MSC/04/17).

## Statistical analysis

Data were expressed as mean ± standard deviation (SD). Independent samples t-test (at p <0.05) were used to determine statistically significant differences. All analyses were done by using GraphPad Prism software (version 7).

## Results

### In vitro targeting of HPV 16E7 inhibited cell proliferation via Rb protein restoration

We first established if the targeting of HPV 16E7 with CRISPR/Cas9 in the mouse HPV transformed cell line, TC-1, would result in changes in cell growth. It was observed that a significant reduction in cell proliferation occurred with 16E7-targeting gRNA and that this effect was specific to HPV 16 +ve cell lines, TC1 and CasKi, while HPV -ve C33A cells were not affected (Fig 1A). Consistent with our previous work [6], transfection with a nonspecific gRNA or Cas9-expressing plasmids alone had a small but reproducible effect, likely due to nonspecific DNA toxicity (800 ng/well, 24-well plate). We next assessed the long-term effect of treatment using colony-forming assays. The results mirrored the viability assays, with a significant reduction in the number of colonies in HPV16 +ve cell lines compared to other controls (Fig 1B), indicating highly specific killing of 16E7-expressing cells. Because E7 protein binds to Rb protein resulting in its degradation [17], we assessed Rb expression levels by western blotting and showed

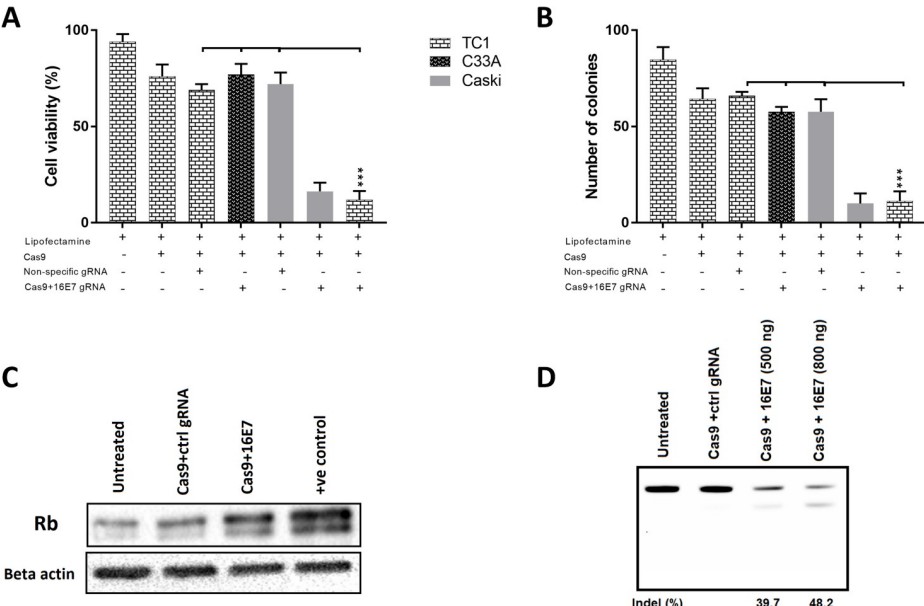

**Fig 1. The targeting of HPV 16E7 gene with CRISPR/Cas9 inhibited cell proliferation and restored Rb protein expression.** (A) HPV 16 +ve (TC1 and CasKi) and HPV -ve (C33A) cell lines were treated with Cas9+ 16E7 specific gRNA or control gRNA (nonspecific) for 72 hours before cell viability was determined by MTT assay. (B) HPV 16 +ve (TC1 and CasKi) and HPV -ve (C33A) cell lines were treated with Cas9+ 16E7 or control gRNAs and allowed to form colonies for two weeks, then the number of colonies was counted. (C) TC1 cells were treated with either Cas9+16E7 or Cas9+ control gRNAs for 72 hours before Rb protein expression was determined by western blot. Jurkat lysate was used as a positive control. Beta-actin was used as loading control. (D) TC1 cells were treated with Cas9+16E7 or control gRNAs for 72 hours, then editing efficiency was determined by T7E1 assay. Two DNA concentrations (500 ng or 800 ng per well, 24-well plate) were tested. All data are presented as mean ± SD. Statistical difference was assessed by ANOVA with post-hoc analysis, $^*p<0.05$, $^{**}p<0.01$, $^{***}p<0.001$.

that it was higher (Fig 1C) in 16E7 gRNA-treated cells, consistent with a knockout of HPV 16E7 gene expression. Finally, we investigated the editing efficiency of 16E7 gene in TC1 cells using the T7E1 assay, showing that 48.2% of the target gene was successfully edited through the non-homologous end joining repair pathway (NHEJ), when 800 ng of DNA/well were transfected (24-well plate). Increasing DNA concentration beyond that has shown to be toxic.

## The systemic delivery of Cas9/16E7 targeting gRNA packaged in stealth liposomes effectively cleared tumour xenografts in immunocompetent mouse model with minimal inflammation

We and others have previously shown the effects of CRISPR/Cas in cancer models that use xenografts in immunodeficient mice [6,18,19]. This misses an important aspect of responses to treatment, namely the immune response itself. To address this, we tested the *in vivo* efficacy of targeting 16E7 in an immunocompetent mouse model. Cas9 plasmids and gRNAs were packed in stealth liposomes which we have previously used extensively with siRNA *in vivo* work [10,16] and injected into the tail vein of mice that had established TC1 tumours. We observed a good control of tumour growth compared to experimental controls in the first arm, however, growth suppression was not maintained; with tumours growing when 16E7 specific treatment ceased at day 14, and ultimately reaching the experimental endpoint (1000 mm³) by day 36 (Fig 2A). The second treatment arm, which continued to receive 16E7 specific treatment, showed a complete growth inhibition by the end of the experiment, and tumours were effectively cleared (Fig 2A). All controls reached the endpoint by day 22.

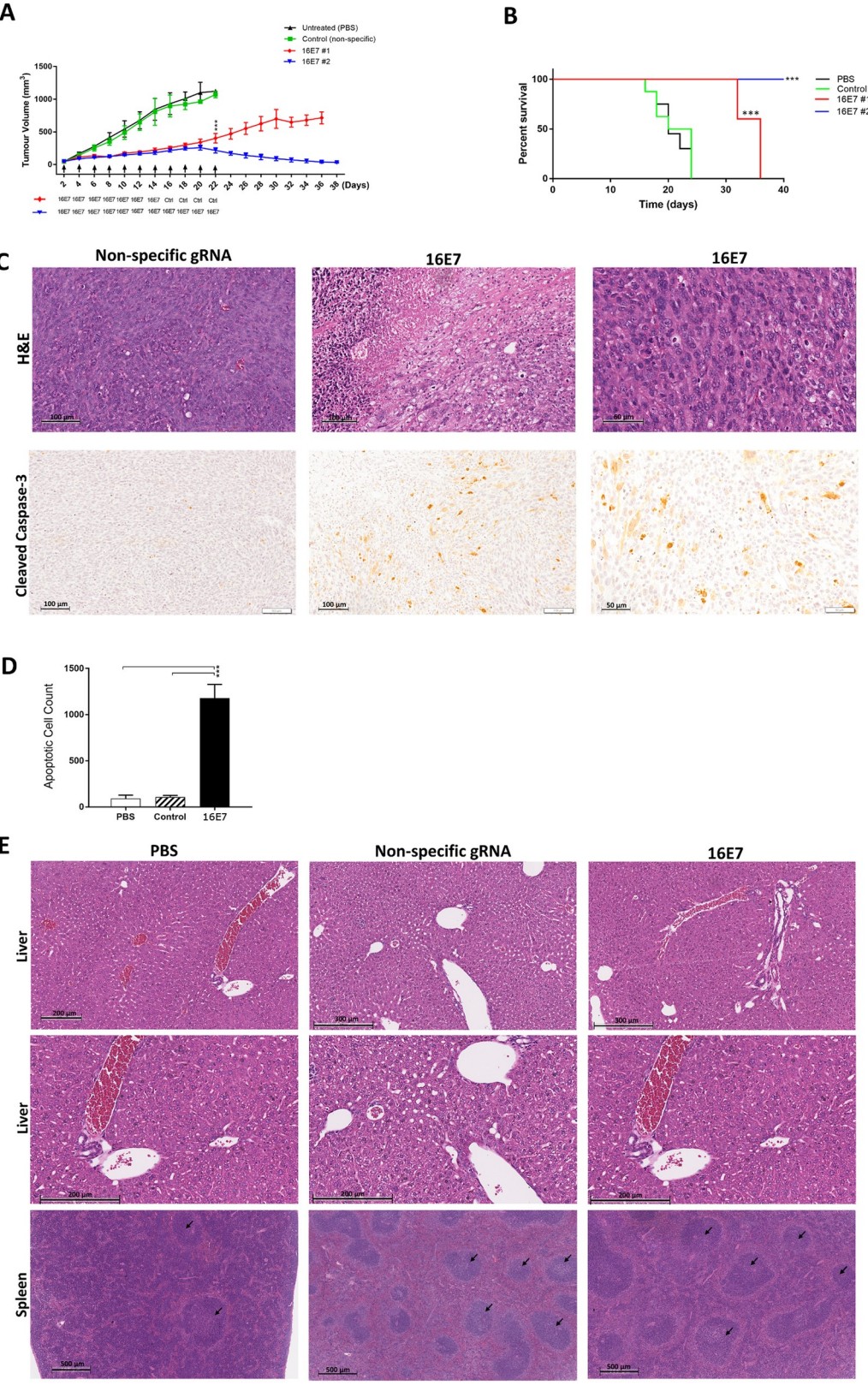

**Fig 2. The systemic administration of Cas9/16E7 targeting gRNAs coated in PEGylated liposomes effectively cleared established HPV 16-driven tumours via apoptosis with minimal toxicity.** (A) TC1 (HPV 16 +ve) cells were

subcutaneously inoculated in C57BL/6J and allowed to grow to $\approx 50$ mm³ before treatments were injected via tail vein (a total of 10 µg of plasmid DNA expressing Cas9 and 16E7 targeting gRNA for treatment groups, or Cas9 and nonspecific gRNA for control, or PBS for untreated group). Injections were administered second daily. First treatment arm (16E7#1) received a total of seven 16E7 treatments, then another four control (Cas9+nonspecific gRNA) injections. The second treatment arm (16E7#2) received a total of eleven 16E7 injections. Tumour volume was measured second daily with digital caliper. The experiment endpoint was tumour volume reaching 1000 mm³. (B) The survival analysis of established TC1 xenografts after 16E7 targeting, with similar experimental groups as in A. (C) Immunohistochemical staining of tumour specimens with H&E staining (upper panel) or cleaved caspase-3 (lower panel) for control (nonspecific) or 16E7 treated mice. (D) The apoptotic cell counts in cleaved caspase-3 stained tumour samples from untreated, control, or 16E7 treated mice. (E) H&E staining of liver and spleen specimens from untreated, control, or 16E7 treated mice. N = 6 per group. Data were represented as mean ± SD. Statistical significance was assessed by ANOVA with post-hoc analysis. * p<0.05, ** p<0.01, *** p<0.001.

This finding confirmed the specificity of treatment to HPV 16E7-driven tumours, and that its effect is independent of a possible "empty vector effect" as reported elsewhere [20–22]. Targeting 16E7 gene with seven doses prolonged cancer-free survival by 12 days (36 versus 24 days, for 16E7#1 vs control, respectively, P<0.001), while eleven consecutive 16E7 doses (the second treatment arm) eliminated tumour entirely (Fig 2B). H&E staining of tumour specimens (from treatment group #1) showed a markedly increased number of apoptotic cells in 16E7 treated tumours, with degenerative cells and extensive necrotic regions (Fig 2C). Staining for cleaved caspase-3 showed 11-fold increase in apoptosis in 16E7 treated mice compared to the control (Fig 2C and 2D).

Intravenously injected lipoplexes were previously shown to accumulate largely in the liver and spleen [23,24]. Some evidence suggests that DOTAP in certain isoforms could activate CD8[+] T-cells mediated immune response through the production of reactive oxygen species [25,26]. We previously investigated our HFDM liposomes loaded with siRNA and showed that they did not significantly induce pro-inflammatory cytokines [IFN α, IL-6, IFN γ) [27,28]. To further investigate this, we performed H&E staining of the liver, spleen and tumour specimens from mice treated with PBS only, or HFDM liposomes loaded with CRISPR/Cas9 (Fig 2E). There was no evidence of significant inflammation or necrosis, with minimal leukocytic recruitment in liver specimens from mice treated with HFDM liposomes compared to the untreated group. On the other hand, splenic reactive follicular hyperplasia was noticed in the control gRNA and treated mice compared to the untreated murine spleen, possibly a nonspecific and inevitable reaction to plasmid DNA due to its high frequency of CpG motifs, a known feature of bacterial DNA [29,30].

## The CRISPR/Cas9 mediated cell death is not immunogenic

To assess whether the observed cell death was also immunogenic, we examined H&E stained tumour specimens for infiltration of inflammatory cells (Fig 3A). It showed extensive necrotic regions peripherally, which is expected with the systemic delivery of the treatment. Apoptotic and degenerative changes were also noticed, but no evidence of significant inflammation within tumour tissue was detected. Our data support the extracellular release of ATP, with a significant increase 72 hours after 16E7 treatment (Fig 3B). The western blot analysis of both adherent cells and media showed that the HMGB-1 protein was passively released in the media of 16E7 treated cells 72 hours after treatment (Fig 3C).

Next, we examined if treating TC1 cells would induce an immune response in immunocompetent mice. To ensure the pretreated TC1 cells were viable at the time of inoculation to induce antitumour immunity, the effect of treatment on cell viability was assessed at different time points (Fig 4A). It was shown that 48 hours post-treatment was an ideal time point to prepare cells for subcutenous injection (Fig 4B). The *ex vivo* treatment with 16E7 appeared less

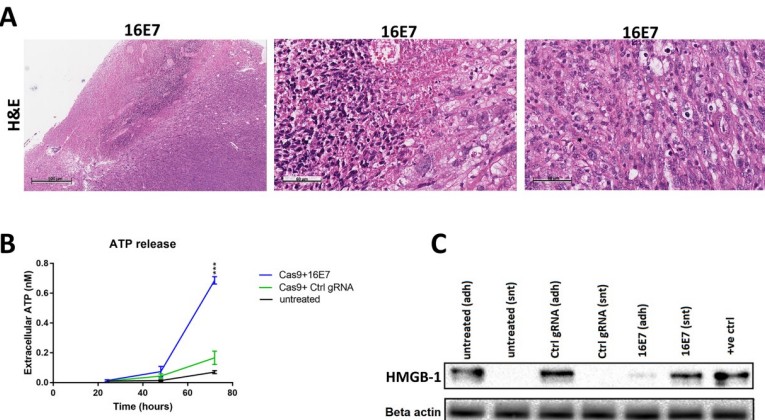

**Fig 3. CRISPR/Cas9-induced cell death in HPV 16-driven xenografts is not immunogenic.** (A) Examination of H&E stained tumour specimens from Cas9+16E7 treated mice (16E7 #1 group) for inflammatory cells infiltration. (B) Extracellular ATP release assay of TC1 cells (HPV 16+ve) treated with Cas9+16E7, Cas9+nonspecific gRNA, or untreated, over 72 hours. A total of 800 ng of plasmid DNA was transfected per well (24-well plate) at 70% cell confluency. (C) TC1 cells were treated with Cas9+16E7, Cas9+nonspecific gRNA (control), or untreated for 72 hours before protein release of HMGB-1 was assessed by western blot analysis. Samples were collected from adherent cells (adh) or supernatant (snt) for each group. HeLa cells lysate was used as positive control. Data were represented as mean ± SD. Statistical significance was assessed by ANOVA with post-hoc analysis. * $p < 0.05$, ** $p < 0.01$, *** $p < 0.001$.

effective compared to cisplatin and mitoxantrone (Fig 4C, left flank). However, this could result from the less efficient transfection of plasmid DNA compared to other controls. When rechallenged with viable TC1 cells, 16E7 treatment did not seem to induce anticancer immunity in the host, and thus tumour growth rate was not significantly different from cisplatin-treated mice (Fig 4C, challenge flank), with no survival advantage (Fig 4D). Therefore, we conclude that the targeting of E7 with CRISPR/Cas9, while inducing apoptosis in target cells, does not result in immunogenic cell death.

## Discussion

It is now clear that a major challenge to developing CRISPR/Cas9 therapeutics in humans is the delivery of the gene-editing components to target organs with acceptable safety profiles and minimal immunogenic/antigenic toxicity. After a decade of clinical trials, the use of PEGylated liposomes has shown promising results, particularly in the anticancer gene therapy area [31]. Here we demonstrate that the intravenously administered Cas9/16E7sgRNA plasmids packaged in stealth liposomes effectively cleared established 16E7-driven tumours in syngeneic mice with a significant survival advantage. These results are comparable to efficacy testing in immunocompromised mice [6]. The effect was specific to 16E7 gene targeting as demonstrated by the two treatment arms, with one arm resuming tumour growth once the 16E7 treatment was ceased. This rules out the "empty vector effect", defined as the nonspecific release of cytokines and activation of NK cells elicited by the immunogenicity of the empty vector itself without the therapeutic genes, resulting in a transient anti-tumour effect [20,22].

To the best of our knowledge, this paper is the first to report on the *in vivo* efficacy and immunogenicity of CRISPR/Cas9 coated in stealth liposomes in an immunocompetent mouse model. Host immune responses to CRISPR/Cas9, delivery vehicle, or gene expression of the edited genes have been reported since the early studies on CRISPR/Cas9 [32]. Both humoral and T cell-mediated immunity against *Streptococcus pyogenes* and Cas9 antibody were detected in more than 80% of healthy individuals [33–35]. We did not measure the anti-Cas9 antibody in the blood stream after treatment, and thus we cannot claim an immune response did not

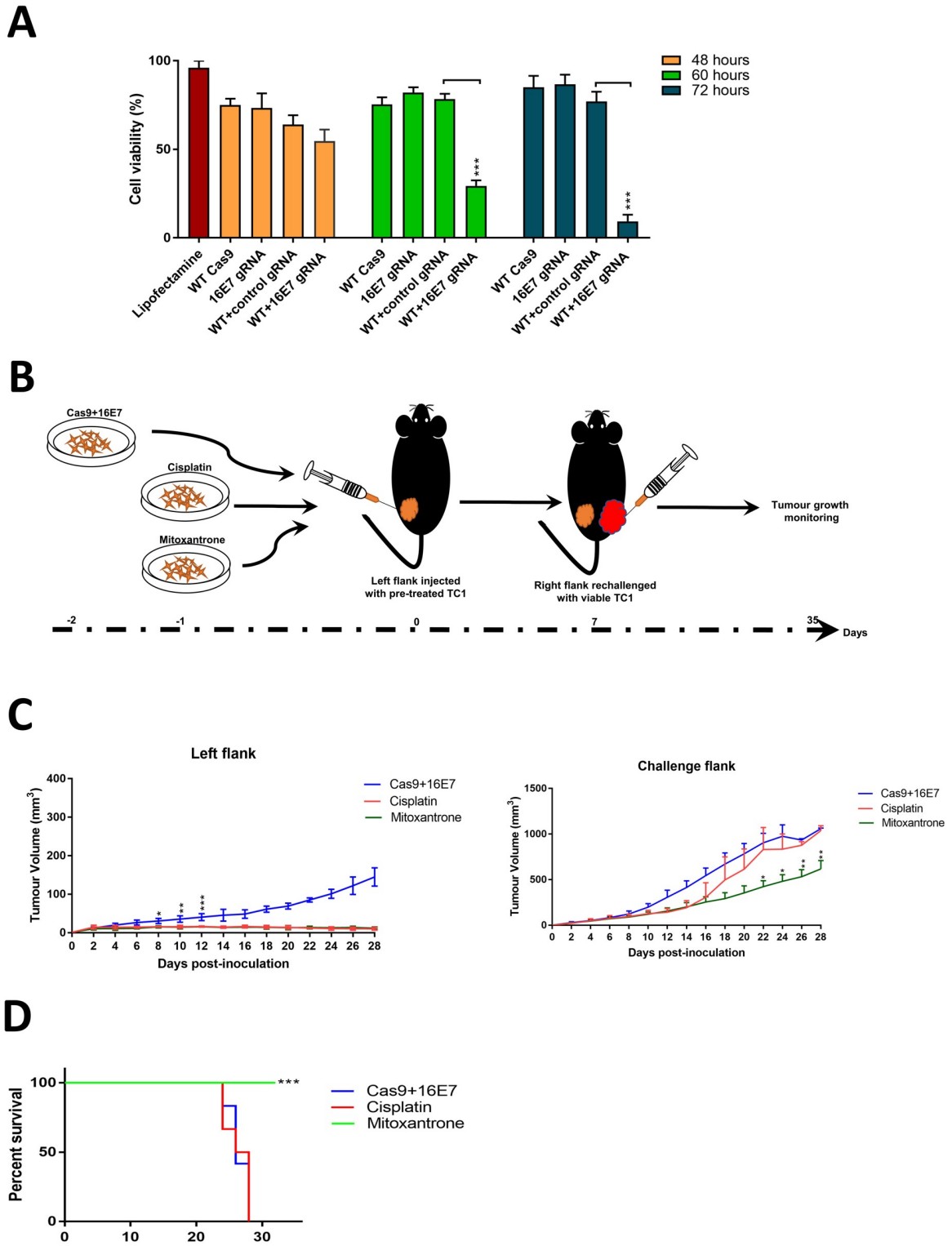

**Fig 4. *In vivo* testing of the immunogenicity of cell death induced by CRISPR/Cas9 targeting HPV 16-driven tumours.** (A) Cell viability assay of HPV 16 +ve (TC1) cell line treated with Cas9 only, 16E7 gRNA only, Cas9+control gRNA or Cas9 +16E7 gRNA for 48, 60, or 72 hours before viability was assessed by MTT assay. (B) The experiment design and timeline to assess immunogenic cell death. TC1 cells were treated with either Cas9+16E7 (test group), cisplatin (non-ICD inducing, negative control), mitoxantrone (ICD inducing, positive control), injected in the left flank of C57BL/6J mice, and allowed to grow for seven days. The right flank was then rechallenged

with viable (untreated) TC1 cells and followed up to the experiment endpoint (tumour volume of 1000 mm³). Tumour volume was assessed by digital caliper. (C) The tumour volume assessment of pre-treated TC1 cells (left flank) or the challenge viable cells (right flank) as explained in Fig 2A. (D) survival analysis of established TC1 xenografts after exposure to pre-treated TC1 cells with either 16E7, cisplatin, or mitoxantrone (right flank tumours). N = 5 per group. Data were represented as mean ± SD. Statistical significance was assessed by ANOVA with post-hoc analysis. * p<0.05, ** p<0.01, *** p<0.001.

occur. However, it did not seem to affect *in vivo* efficacy of the treatment or the wellbeing of the mice. We previously demonstrated that HFDM liposomes could protect siRNA payloads from circulating reticuloendothelial cells with no significant activation of the immune system [10], which could explain the consistent *in vivo* efficacy of targeting the 16E7 gene. Moreover, administering the lowest efficacious dose of the treatment was previously suggested to minimize the chance of developing anti-Cas9 immune response [32]. Here we administered small but more frequent treatments (10 µg per dose, 11 total treatments) instead of the widely adopted regimen of 30–60 µg per dose (2 mg/kg). Higher doses of CRISPR/Cas9 therapeutics correlates with long-term expression of Cas9, which could potentially trigger an immune response [32].

We argue that our HFDM liposomes effectively delivered payloads to target organs as a result of optimal characterization of key determinants for a successful delivery vehicle. Coating liposomes with PEGylation was shown to increase its stability, reduce renal clearance and prolong circulation time [36]. PEGylation also increased particle size, limiting the extravasation of liposomes into healthy tissues with intact endothelium, while preferentially accumulated in tumour tissues via the enhanced permeation and retention effect [37], leading to an improved transfection efficiency. Using DOPE is also implicated in the enhanced intracellular processing of plasmid DNA. DOPE was shown to play an important role in destabilizing endosomal membrane due to its fusogenic properties under acidic conditions, resulting in the release of DNA into the cytoplasm [38]. Another pertinent aspect of HFDM liposome efficiency was its particle size and homogeneity (N/P ratio = 16:1, average particle diameter = 217 nm ± 13.35, average PDI = 0.39 ± 0.04, zeta-potential = +48 ±3.12mV). Lipoplexes with smaller particle size (<200 nm) were shown to be mainly internalized via clathrin-mediated endocytosis, and thus rapidly cleared by lysosomal compartment. On the other hand, larger lipoplexes were internalized by the kinetically slow caveolea-mediated endocytosis, allowing for cytoplasmic escape of cargo [39], and thus enabled more efficient intracellular processing. We previously showed that stealth liposomes prepared with HFDM displayed less hepatic uptake [16], with no significant increase in lactate dehydrogenase after IV treatment of mice [27,28], which explains the observed minimal changes in the liver of treated mice. Of note, the reactive splenic follicular hyperplasia observed in our experiment was not surprising as it was previously shown that the administration of fluorescently labelled lipids in mice did not leave vasculature except in the spleen due to its discontinuous endothelium [40]. Plasmid DNA derived from bacteria differs from eukaryotic DNA by the high frequency of hypomethylated CpG motifs, which induces B-cells and pro-inflammatory cytokines [29]. Hence, it is possible to reduce potential splenic toxicity by methylating these motifs prior to systemic administration [20,21]. Our data also suggest that the apoptotic cell death induced by CRISPR/Cas9 was not immunogenic, as the rechallenge experiment failed to establish the presence of an immune response against HPV 16E7-driven TC1 cells.

Altogether, our data show that HFDM liposomes are a promising delivery system for CRISPR therapeutics, with high *in vivo* efficacy in syngeneic mice. The systemic administration of CRISR-lipoplexes showed minimal hepatic toxicity and did not induce inflammation. Although the observed cell death was apoptotic but not immunogenic, other strategies may be explored to enhance the host anticancer immunity [41].

## Supporting information

**S1 Raw images.**
(PDF)

## Author Contributions

**Conceptualization:** Nigel A. J. McMillan.

**Data curation:** Luqman Jubair, Sora Fallaha.

**Formal analysis:** Luqman Jubair, Alfred K. Lam.

**Funding acquisition:** Nigel A. J. McMillan.

**Investigation:** Luqman Jubair, Sora Fallaha.

**Methodology:** Luqman Jubair.

**Project administration:** Nigel A. J. McMillan.

**Resources:** Nigel A. J. McMillan.

**Supervision:** Nigel A. J. McMillan.

**Writing – original draft:** Luqman Jubair.

**Writing – review & editing:** Sora Fallaha, Nigel A. J. McMillan.

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
