## [Decision Letter · Decision Letter 0]

9 Oct 2019

PONE-D-19-24737

CRISPR/Cas9-loaded stealth liposomes effectively clear established HPV16-driven tumours in syngeneic mice.

PLOS ONE

Dear Professor McMillan,

Thank you for submitting your manuscript to PLOS ONE. After careful consideration, we feel that it has merit but does not fully meet PLOS ONE’s publication criteria as it currently stands. Therefore, we invite you to submit a revised version of the manuscript that includes a point-by-point  answer to all the revieweres' comments.    The reviewers think that the authors need to describe the synthesis method of the liposomes that they used in the study and the characterization of  their physico-chemical properties. In addition, successful delivery of the liposomes/Cas9toin the tumor site should be shown. A complete account of the comments can be found at the end of this letter.

We would appreciate receiving your revised manuscript  by Nov 23 2019 11:59PM.  To enhance the reproducibility of your results, we recommend that if applicable you deposit your laboratory protocols in protocols.io, where a protocol can be assigned its own identifier (DOI) such that it can be cited independently in the future. For instructions see: http://journals.plos.org/plosone/s/submission-guidelines#loc-laboratory-protocols

We look forward to receiving your revised manuscript.

Kind regards,

Valentin Ceña

Academic Editor

PLOS ONE

Journal Requirements:

2. Please provide additional information about each of the cell lines used in this work, including source, history, culture conditions and any quality control testing procedures (authentication, characterisation, and mycoplasma testing). For more information, please see " ext-link-type="uri" xlink:type="simple">http://journals.plos.org/plosone/s/submission-guidelines#loc-cell-lines."

3. At this time, we request that you  please report additional details in your Methods section regarding animal care, as per our editorial guidelines:

(1) Please provide details of animal welfare (e.g., shelter, food, water, environmental enrichment)

(2) Please include the secondary method of euthanasia in addition to carbon dioxide asphyxiation

(3) Please state the duration of the experiment

(4) Please state the frequency of monitoring of the mice after tumor cell inoculation

Thank you for your attention to these requests

'The funders had no role in study design, data collection and analysis, decision to

publish, or preparation of the manuscript.'

Please provide an amended Funding Statement that declares *all* the funding or sources of support received during this specific study (whether external or internal to your organization) as detailed online in our guide for authors at http://journals.plos.org/plosone/s/submit-now.  Please state what role the funders took in the study.  If any authors received a salary from any of your funders, please state which authors and which funder. If the funders had no role, please state: "The funders had no role in study design, data collection and analysis, decision to publish, or preparation of the manuscript."

Additional Editor Comments (if provided):

Reviewers' comments:

Reviewer's Responses to Questions

**Comments to the Author**

1. Is the manuscript technically sound, and do the data support the conclusions?

Reviewer #1: Yes

Reviewer #2: Yes

2. Has the statistical analysis been performed appropriately and rigorously? 

Reviewer #1: Yes

Reviewer #2: Yes

3. Have the authors made all data underlying the findings in their manuscript fully available?

Reviewer #1: Yes

Reviewer #2: Yes

4. Is the manuscript presented in an intelligible fashion and written in standard English?

Reviewer #1: Yes

Reviewer #2: Yes

5. Review Comments to the Author

Reviewer #1: Jubair L et al demonstrated the power of CRISPR/Cas9 to inhibit the expression of 16E7 and the antitumoral implications of these both in vitro and in vivo in an immunocompetent tumor model. The nano-delivery of gRNA guides targeting 16E7 show a potent anti-tumoral effect without obvious toxicity in major organs. The results show apoptosis induction in the tumor biopsies and while absence of immunogenic cell death.

The paper shows a strong anti-tumoral effect in vivo in a syngeneic tumor model which is quite interesting. However, there are a number of major points that the authors need to address. For example, the authors did not describe the synthesis method of the liposomes that they used in the study. Also, the physical and chemical characterization of liposomes is lacking such as size, shape and charge. Also, they did not prove the successful delivery of the liposomes/Cas9 in the tumor site. I would suggest authors show evidence of tumoral delivery of Cas9 by IHC or IF.

Other points to address

As the paper is focused on the utilization of in-house liposomes, why did not the authors perform the cell transfections with those liposomes instead of lipofectamine 3000? I think the paper will benefit from additional experiments done with their liposomes or an experiment comparing the in vitro transduction efficiency of lipofectamine versus their liposomes in reducing the expression of 16E7.

In Fig 1A, there is a ~25% reduction in cell viability in TC-1 cells transfected with Cas9 only and Cas9+unspecific gRNA. Although the authors explain that is due to unspecific DNA toxicity and is not significant, it looks significant to me. Can you show p values? Figure 1A also needs the experimental negative controls for C33A and Caski cells (as TC-1 cells).

Similarly, for 1B, do the authors have the negative controls for C33A and Caski colonies? For a better assessment of clonogenicity inhibition, it will also be interesting to show the images of the colony formation assays.

In Figure 2B, the Y axis annotation should be changed as it´s not natural survival of mice bearing tumors but percentage mice not reaching the ethical culling point. The same in 4D.

In the in vivo experiments, how were the tumors measured? What was the formula used to determine the tumor volumes?

It would be good to provide a TUNEL assay of the tumor biopsies or another additional technique that detect apoptotic cell death to further prove existence of apoptosis.

As a non-specialist in immunology or pathology, can the authors point in the HE images the “minimal leukocytic recruitment” and also the “splenic reactive follicular hyperplasia” in the slides presented? Can the authors mention where/how the slides were analyzed? In a pathology service?

Figure 3C lacks a protein loading control for the Western blot.

Line 292, substitute “indeal” for “ideal”.

Reviewer #2: This is a very elegant paper describing well performed experiments and paving the way towards CRISPR/Cas9 treatment in cancer processes.

Three errors shoud be corrected:

Line 75 should say "humoral" instead of "humeral"

line 292 should say "ideal" instead of "indeal"

The inside legend of fig 2D (countings) should say "16E" instead of "18E"

The first group of experiments demonstrate that in vitro targeting of HPV 16E7 with CRISPR/Cas9 in the mouse 180 HPV transformed cell line, TC-1, would results in changes in cell growth, inhibiting proliferation, via enhanced expression of Rb protein. The authors mention that this occurs with 16E7-targeting gRNA and that this effect was specific to HPV 16 +ve cell lines, TC1 and CasKi, while HPV -ve C33A cells were not affected. ¿Could the authors mention this again in the discussion and explain why this last line had different behaviour?

The second group of experiments demonstrate that the systemic delivery of Cas9/16E7 targeting gRNA packaged in stealth liposomes effectively clears tumour xenografts in immunocompetent mouse model with minimal inflammation via apoptosis. The countings of cleaved-caspase 3 in fig 2D however do not reflect well what is shown in image 2C right hand side. One would expect to see many more immunostained cells. I suggest showing a better photomicrograph, to document the countings..

6. PLOS authors have the option to publish the peer review history of their article (what does this mean?). If published, this will include your full peer review and any attached files.

Reviewer #1: No

Reviewer #2: No

---

## [Author Response · Author response to Decision Letter 0]

27 Oct 2020

In uploaded file but copied here too

Response to reviewers (in red): 

Reviewer #1: Jubair L et al demonstrated the power of CRISPR/Cas9 to inhibit the expression of 16E7 and the antitumoral implications of these both in vitro and in vivo in an immunocompetent tumor model. The nano-delivery of gRNA guides targeting 16E7 show a potent anti-tumoral effect without obvious toxicity in major organs. The results show apoptosis induction in the tumor biopsies and while absence of immunogenic cell death.

The paper shows a strong anti-tumoral effect in vivo in a syngeneic tumor model which is quite interesting. However, there are a number of major points that the authors need to address. For example, the authors did not describe the synthesis method of the liposomes that they used in the study. Also, the physical and chemical characterization of liposomes is lacking such as size, shape and charge. Also, they did not prove the successful delivery of the liposomes/Cas9 in the tumor site. I would suggest authors show evidence of tumoral delivery of Cas9 by IHC or IF.

- the protocol for the synthesis of liposomes has been explained in depth in our previous publication (Wu SH et al, 2009), which was cited here (reference number 16). The synthesis process is complex and would be of limited value if summarized in one paragraph, and therefore the authors opted to cite the original publication which details every step of the preparation. The characteristics of the nanoparticles including N/P ration, size, PDI, and charge were mentioned in discussion section (line 357-360). 

While it would be helpful to show the successful delivery of liposomes in the tumour site, it has proven challenging with IHC anti-Cas9 antibody staining (poor affinity and high cross-reactivity). To show the successful delivery of payloads, mCherry or GFP-tagged Cas9 would be ideal to serve this purpose. However, we have previously shown that our PEGylated liposomes were successful to deliver payloads to tumour site after 24 hours of IV injection (Wu SH et al, 2009). In this work, we have shown that the 16E7 targeting consistently halted tumour growth in vivo, and that effect was diminished once the treatment was substituted with control gRNA. 

Other points to address

As the paper is focused on the utilization of in-house liposomes, why did not the authors perform the cell transfections with those liposomes instead of lipofectamine 3000? I think the paper will benefit from additional experiments done with their liposomes or an experiment comparing the in vitro transduction efficiency of lipofectamine versus their liposomes in reducing the expression of 16E7.

- we would expect a lower transfection efficiency of pegylated liposomes compared to lipofectamine for in vitro testing due to the effect of PEGylation, which would hamper the uptake of the DNA, and therefore, we opted to use lipofectamine 3000. Also, comparing the efficiency of the two transfection methods was outside the scope of this work. 

In Fig 1A, there is a ~25% reduction in cell viability in TC-1 cells transfected with Cas9 only and Cas9+unspecific gRNA. Although the authors explain that is due to unspecific DNA toxicity and is not significant, it looks significant to me. Can you show p values? 

- We agree that the effect was marginally significant at (p = 0.047), compared to P0.001 when treated with 16E7 specific treatment. The statement was amended (line 183). 

Figure 1A also needs the experimental negative controls for C33A and Caski cells (as TC-1 cells). Similarly, for 1B, do the authors have the negative controls for C33A and Caski colonies? For a better assessment of clonogenicity inhibition, it will also be interesting to show the images of the colony formation assays.

 - the effect of 16E7 targeting in CasKi cell line was previously published in (Jubair et al 2019), reference number 6, and therefore we only showed the 16E7 treatment group. For the HPV -ve (C33A) cell line, it was included as a biological control and thus we showed the full treatment had no significant effect on its viability. We did not include all the controls for the negative control as it can get too confusing to read. The full controls for C33A cell line can be added as a supplementary figure if required. The same applies to fig 1B. 

In Figure 2B, the Y axis annotation should be changed as it´s not natural survival of mice bearing tumors but percentage mice not reaching the ethical culling point. The same in 4D.

- the definition of survival was clarified in the in vivo testing section as tumours reaching 1000 mm3 to be culled, and therefore it reflects the number of mice with tumours smaller than this cut-off. Percent survival, or survival (%), is therefore widely accepted in medical literature. 

In the in vivo experiments, how were the tumors measured? What was the formula used to determine the tumor volumes?

- tumour volume was measured with digital caliper and the volume was calculated using this formula: volume= 1/2 x L x W x H, where L: length, W: width, H: height. 

This was done according to a published analysis of various formulas and methods (Mary M et al, 1989)

It would be good to provide a TUNEL assay of the tumor biopsies or another additional technique that detect apoptotic cell death to further prove existence of apoptosis.

- There are many methods to detect apoptosis. In our study, we used cleaved-Caspase 3 as caspases are crucial mediators of apoptosis, particularly caspase-3 which catalyzes the cleavage of many key cellular proteins. Therefore, it is a very reliable marker for apoptosis, particularly in cancer cell lines. On the other hand, TUNEL assay detects apoptosis by labelling the 3’-hydroxyl termini in the double-stranded breaks generated during apoptosis. In our work, we used CRISPR to generate double-stranded breaks, which we hypothesized to have caused cell death via apoptosis. If TUNEL assay is used to measure apoptosis, it is possible it may overestimate apoptosis by quantifying any CRISPR induced DNA damage toward apoptotic cell death, regardless of the actual mechanism of death. 

As a non-specialist in immunology or pathology, can the authors point in the HE images the “minimal leukocytic recruitment” and also the “splenic reactive follicular hyperplasia” in the slides presented? Can the authors mention where/how the slides were analyzed? In a pathology service?

- The slides were processed and stained by pathology department at Gold Coast University Hospital, and analysed by senior pathologist, Dr Alfred Lam (co-authored this work). The slides were read and analysed using OlyVIA software. Arrows were added to show the reactive follicular hyperplasia. 

Figure 3C lacks a protein loading control for the Western blot.

- protein loading added 

Line 292, substitute “indeal” for “ideal”.

- corrected

Reviewer #2: This is a very elegant paper describing well performed experiments and paving the way towards CRISPR/Cas9 treatment in cancer processes.

Three errors shoud be corrected:

Line 75 should say "humoral" instead of "humeral"

-corrected 

line 292 should say "ideal" instead of "indeal"

-corrected

The inside legend of fig 2D (countings) should say "16E" instead of "18E"

-corrected

The first group of experiments demonstrate that in vitro targeting of HPV 16E7 with CRISPR/Cas9 in the mouse 180 HPV transformed cell line, TC-1, would results in changes in cell growth, inhibiting proliferation, via enhanced expression of Rb protein. The authors mention that this occurs with 16E7-targeting gRNA and that this effect was specific to HPV 16 +ve cell lines, TC1 and CasKi, while HPV -ve C33A cells were not affected. ¿Could the authors mention this again in the discussion and explain why this last line had different behaviour?

- in our work, we targeted 16E7 gene which is crucial for the survival of cancer cell. In HPV +ve cell lines (TC1, CasKi), the treatment knocked out the expression of this gene, and thus resulted in cell death. On the other hand, C33A is HPV -ve, and thus does not rely on 16E7 gene for survival, and therefore the treatment had no effect on cell viability. This proves the specificity of targeting to HPV 16E7 gene. 

The second group of experiments demonstrate that the systemic delivery of Cas9/16E7 targeting gRNA packaged in stealth liposomes effectively clears tumour xenografts in immunocompetent mouse model with minimal inflammation via apoptosis. The countings of cleaved-caspase 3 in fig 2D however do not reflect well what is shown in image 2C right hand side. One would expect to see many more immunostained cells. I suggest showing a better photomicrograph, to document the countings..

- To improve the accuracy of counting, the number of apoptotic cells was counted using ImageJ software that screened the entire tumour specimen. Given that the tumour specimens were large, low magnification would be required to show more apoptotic cells, however, that would affect the quality of the image (less clear apoptotic cells at very low magnification). Nevertheless, the presented shots clearly show significant difference in the number of apoptotic cells even at high magnification (50 micro).

---

## [Decision Letter · Decision Letter 1]

2 Nov 2020

CRISPR/Cas9-loaded stealth liposomes effectively clear established HPV16-driven tumours in syngeneic mice.

PONE-D-19-24737R1

Dear Dr. McMillan,

We’re pleased to inform you that your manuscript has been judged scientifically suitable for publication and will be formally accepted for publication once it meets all outstanding technical requirements.

Kind regards,

Valentin Ceña

Academic Editor

PLOS ONE

Additional Editor Comments (optional):

Reviewers' comments:

Reviewer's Responses to Questions

**Comments to the Author**

1. If the authors have adequately addressed your comments raised in a previous round of review and you feel that this manuscript is now acceptable for publication, you may indicate that here to bypass the “Comments to the Author” section, enter your conflict of interest statement in the “Confidential to Editor” section, and submit your "Accept" recommendation.

Reviewer #1: (No Response)

Reviewer #2: All comments have been addressed

2. Is the manuscript technically sound, and do the data support the conclusions?

Reviewer #1: Yes

Reviewer #2: (No Response)

3. Has the statistical analysis been performed appropriately and rigorously? 

Reviewer #1: Yes

Reviewer #2: (No Response)

4. Have the authors made all data underlying the findings in their manuscript fully available?

Reviewer #1: Yes

Reviewer #2: (No Response)

5. Is the manuscript presented in an intelligible fashion and written in standard English?

Reviewer #1: Yes

Reviewer #2: (No Response)

6. Review Comments to the Author

Reviewer #1: (No Response)

Reviewer #2: (No Response)

7. PLOS authors have the option to publish the peer review history of their article (what does this mean?). If published, this will include your full peer review and any attached files.

Reviewer #1: No

Reviewer #2: No

---

## [Editor Report · Acceptance letter]

29 Dec 2020

PONE-D-19-24737R1 

CRISPR/Cas9-loaded stealth liposomes effectively cleared established HPV16-driven tumours in syngeneic mice. 

Dear Dr. McMillan:

I'm pleased to inform you that your manuscript has been deemed suitable for publication in PLOS ONE. Congratulations! Your manuscript is now with our production department. 

Kind regards, 

on behalf of

Dr. Valentin Ceña 

Academic Editor

PLOS ONE